# Smart Cities and Citizen Adoption: Exploring Tourist Digital Maturity for Personalizing Recommendations

Gabriel Marín Díaz [1,*] , José Luis Galdón Salvador [2] and José Javier Galán Hernández [1]

1   Faculty of Statistics, Universidad Complutense, Puerta de Hierro, 28040 Madrid, Spain; josejgal@ucm.es
2   Management Department, Universitat Politècnica de València, Camí de Vera s/n, 46022 Valencia, Spain; jogalsal@doe.upv.es
*   Correspondence: gmarin03@ucm.es

**Abstract:** Due to the irruption of new technologies in cities such as mobile applications, geographic information systems, internet of things (IoT), Big Data, or artificial intelligence (AI), new approaches to citizen management are being developed. The primary goal is to adapt citizen services to this evolving technological environment, thereby enhancing the overall urban experience. These new services can enable city governments and businesses to offer their citizens a truly immersive experience that facilitates their day-to-day lives and ultimately improves their standard of living. In this arena, it is important to emphasize that all investments in infrastructure and technological developments in Smart Cities will be wasted if the citizens for whom they have been created eventually do not use them for whatever reason. To avoid these kinds of problems, the citizens' level of adaptation to the technologies should be evaluated. However, although much has been studied about new technological developments, studies to validate the actual impact and user acceptance of these technological models are much more limited. This work endeavors to address this deficiency by presenting a new model of personalized recommendations based in the technology acceptance model (TAM). To achieve the goal, this research introduces an assessment system for tourists' digital maturity level (DMT) that combines a fuzzy 2-tuple linguistic model and the analytic hierarchy process (AHP). This approach aims to prioritize and personalize the connection and communication between tourists and Smart Cities based on the digital maturity level of the tourist. The results have shown a significant correlation between technology usage and the potential for personalized experiences in the context of tourism and Smart Cities.

**Keywords:** Smart Cities; TAM; decision making; fuzzy logic; Smart Tourism; Customer Journey

## 1. Introduction

In recent years, the world has witnessed constant and rapid changes occurring in an environment that has become increasingly volatile, uncertain, complex, and ambiguous. This fact combined with the economic growth and the emergence of new technologies are bringing new opportunities to the cities and attracting more and more people from the countryside to the city [1].

The concept of Smart Cities has evolved through three generations [2]. Smart City 1.0, driven by technology giants such as IBM and Cisco, offered urban areas technological solutions to manage challenges efficiently. Smart City 2.0 shifted the focus to enhancing the quality of life using technology, while still maintaining control over its implementation. This generation addressed social challenges, citizen well-being, and public services, considering factors beyond technology. In Smart City 3.0, citizens gained prominence in addressing urban issues, supporting city managers in finding practical solutions for social, environmental, and governmental challenges. Emphasizing social inclusivity, this generation recognized the importance of equal participation. It acknowledged that smart solutions need not solely rely on technology and could address urban topics from a broader

perspective. However, efforts were required to include marginalized groups with limited tech skills or resistance. As climate change concerns rose, the emergence of a fourth generation of Smart Cities sought to align the digital transition with the green transition.

Based on projections from the United Nations, it is anticipated that around 68% of the global populace will reside in urban regions by the year 2050, and in this context, new technologies together with the fact that the needs of the inhabitants are growing, suggest the need for a new concept of city capable of responding to the new environmental, economic and sustainability challenges.

In this context emerges the concept of Smart City, widely studied by the scientific community but without a clear consensus neither in its definition nor in the variables that should be measured to evaluate the degree of "intelligence" of a city [3].

In general terms, Smart Cities can be known as connected cities using technology and innovation to enhance the well-being and living standards of their inhabitants, optimizing the use of resources, and promoting sustainable development. These cities use interconnected infrastructures and information systems to collect real-time data, make informed decisions, and offer efficient services to their citizens [1].

Exploring the existing literature concerning the concept of a Smart City [4], two distinct lines of definition can be identified. The first line of argument defended the idea of a Smart City as a connected urban space linked to new technologies and the extensive use of data [5,6]. The second line, defines the concept of a Smart City through variables such as accessibility, human capital, sustainability, and the importance of new technologies [7]. All these factors play a pivotal role in defining Smart Tourism. While extensively studied, the concepts of Smart City and Smart Tourism are often perceived as distinct and almost separate entities. However, they share numerous commonalities, primarily revolving around the extensive utilization of data and emerging technologies, encompassing similar variables. Attempting to rigidly separate these concepts becomes impractical as everyday life and tourism increasingly intertwine, and smart technologies dissolve the boundaries between residential and tourism zones [8].

Smart Tourism seeks to use technology to offer personalized, efficient, and sustainable services to tourists, as well as to enhance the competitiveness of tourist destinations. This involves the use of different technologies such as mobile applications, geographic information systems, artificial intelligence (AI), internet of things (IoT), and virtual reality, among others. The most important goals of Smart Tourism focus on improving the tourist experience and optimizing the management of tourist destinations [9].

Smart destinations involve technologies used to improve the planning and management of tourist destinations, providing real-time information on traffic conditions, accommodation availability, local events, personalized recommendations, etc. [10]. On the other hand, the tourist experience includes among others, mobile applications to provide personalized information and services during the trip, such as interactive guides, translators, restaurant recommendations, and activity reservations, to better enjoy their Customer Journey [11].

Smart Tourism aims to improve the tourist experience, make destinations more accessible and sustainable, and enhance the competitiveness of tourist destinations through technological innovation. In this vein, the following questions to be considered: Is it enough to use technology to talk about Smart City? And more importantly, is it the same to deploy disruptive technologies in a city and to have the citizens themselves adopt them efficiently?

The acceptance of technology plays a crucial role in the successful implementation and sustainability of Smart Cities and Smart Tourism initiatives [4]. The level of acceptance among citizens, businesses, and government stakeholders can significantly influence how effectively these initiatives are implemented and how long they remain viable. Terms such as citizen engagement, user experience, trust, public perception, integration, and interoperability have become critical to making Smart Tourism initiatives more efficient and sustainable. The main motivation of this paper is to fill this gap in the literature by giving recommendations on how to enhance these initiatives.

Technologies employed in Smart Tourism can contribute significantly to the overall efficiency of a Smart City by enhancing various aspects of urban life. Data-driven decision making, traffic and mobility management, and enhanced public safety are some examples of how Smart Tourism technologies can have a broader impact on the efficiency of Smart Cities [12].

Citizens are frequently viewed merely as users, testers, or consumers, disregarding their potential as creators and contributors to creativity and innovation [13]. It becomes evident that to effectively tackle Smart Tourism, the incorporation of the technology acceptance model (TAM) concept into the strategy is essential.

The TAM concept was first introduced by Davis F.D. in 1989 and today is one of the most cited models affecting the adoption and use of technology with more than 78,000 citations according to Google Scholar [14]. The objective of the TAM is to define and comprehend the user's acceptance of a technological system by considering its cognitive, emotional, and behavioral components [15] (Figure 1).

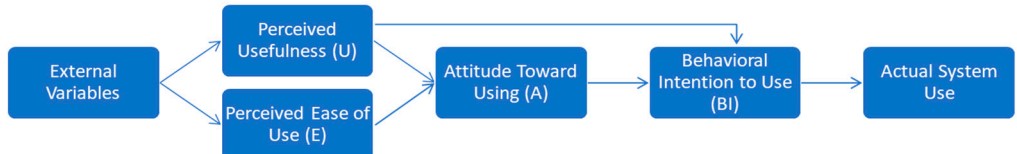

**Figure 1.** Technology acceptance model [14].

Despite the increasing interest in applying TAM, there remains a noticeable lack of systematic applications of the model in the domains of tourism and hospitality. Thus, this study aims to explore the most significant factors or reference points in defining the application of the TAM to tourism. The primary goal is to assess the digital maturity level of tourists concerning their technology usage and subsequently, to develop a system for prioritizing and personalizing the interactions between tourists and tourism agents based on their maturity level. To achieve these objectives, decision-making models based on AHP, information unification within a unified representation domain, and fuzzy logic will be utilized.

Numerous academic works and a vast body of literature exist on the topics of Smart Tourism, Smart Cities, and the TAM. However, there is a noticeable gap in integrating these three concepts, with limited studies exploring potential connections between TAM and enhancing the tourist journey process. Additionally, upon comprehensive analysis, it becomes evident that most of these studies primarily adopt a qualitative approach, leaving an opportunity for future research to employ quantitative methodologies and mathematical models for a more comprehensive investigation.

The model presented in this paper offers several key contributions that deserve attention:

- A methodology is proposed that combines the AHP method and the fuzzy 2-tuple linguistic model to enhance decision-making processes.
- The criteria for the new model are established through an exhaustive review of the literature, where the most frequently cited and relevant factors in the domain are identified and considered.
- A model of the individual prioritization and personalization of interactions is established, according to the tourist's level of digital maturity, and presents a recommendation system, Customer Journey, associated with different types of clusters.

The rest of this article is structured in the following manner: In Section 2, an extensive review of the existing literature related to the fundamental concepts incorporated in the proposed model is conducted. Section 3 presents the detailed methodology employed in this study, outlining the procedures and techniques utilized. In Section 4, a new model designed to calculate TAM is described, building upon the groundwork laid in previous sections. Section 5 illustrates a practical application of the proposed model, demonstrating its effectiveness in a real-life scenario. Finally, in Sections 6–8, the obtained results are

comprehensively discussed, key conclusions are drawn, and any limitations identified in the work are addressed. Moreover, potential avenues for future research and development in this field are outlined.

## 2. Literature Review

### 2.1. Recommender Systems

A recommender system is a type of computer system designed to suggest or recommend items to users, such as products, services, movies, music, news, or any other type of content, based on their preferences, past behaviors, or user profiles. The main goal of a recommender system is to provide personalized and relevant recommendations, to help users discover new items that may be of interest to them [16].

In the literature, different perspectives can be found on the definition of recommender systems. Linked Data comprises a collection of guidelines and practices that facilitate the publication and interlinking of structured data on the Web [17]. This work incorporates human cognitive processes, personality, and affective signals into recommendation models [18]. In general, articles suggest that recommendation systems are computer tools that use machine learning techniques to model and predict user preferences based on behavioral data, and that there are ongoing efforts to improve their personalization, accuracy, and evaluation.

Recommendation systems applied to Smart Cities are tools that use data and algorithms to provide personalized and relevant recommendations in the context of Smart City services and applications. These systems focus on improving the efficiency and quality of life of citizens by providing suggestions and guidance based on their preferences, needs, and the urban environment [19].

Upon investigating the literature related to the search variable TS = (RECOMMENDER SYSTEMS) more than eight thousand publications evaluating this concept were found (Figure 2). The prominence of the field of recommender systems can be underscored by the considerable number of publications generated by academics and practitioners in this domain.

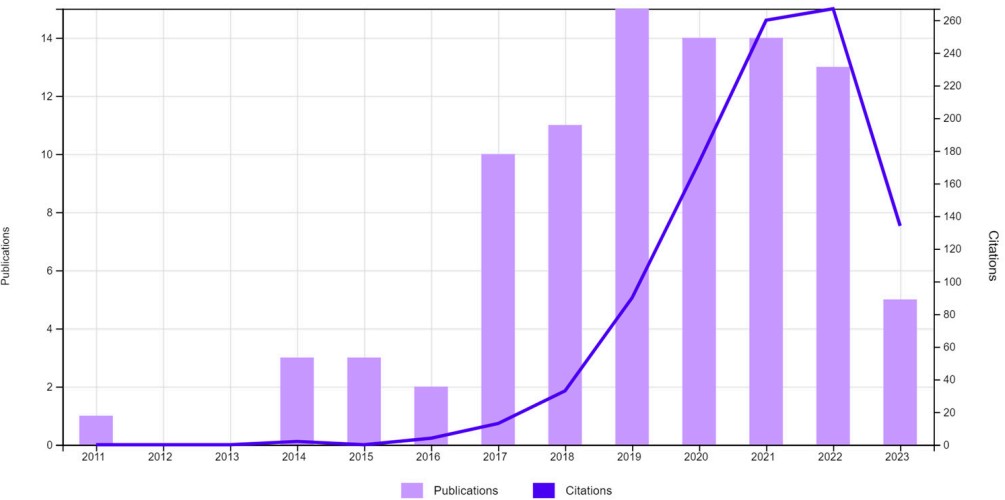

**Figure 2.** Studies related to the recommender systems in the Smart Cities field.

If the focus of the search is on analyzing recommender systems in a Smart City environment (TS = (RECOMMENDER SYSTEMS)) AND (TS = (SMART CITY) OR TS = (SMART CITIES)), only 91 articles are found, indicating a compelling gap worthy of exploration.

The aim is to showcase scientific publications linked to this concept. To accomplish this, research was conducted using the Web of Science Core Collection, employing specific criteria to select journal articles, covering the entire available period.

In the subsequent section, thorough exploration of the concepts of Smart Cities and Smart Tourism will be conducted.

### 2.2. Smart Cities and Smart Tourism

As mentioned in the introduction section, the irruption of new technologies combined with the new challenges of civilization have caused the growth of Smart Cities. In this context, extensive research has been conducted in the domains of Smart Cities.

Smart City technology has a significant impact on the tourism industry. In the following study [20], the strategic role of technology in smart tourist destinations is explored. Another study [21] examines the relationship between Smart Tourism technology and overall satisfaction in three South Korean cities. In this study, [22], it is argued that technology innovations bring stakeholders together in tourism service ecosystems and transform industry structures, processes, and practices. Overall, these papers suggest that Smart City technology can enhance tourism competitiveness and generate new opportunities for the industry.

When conducting research on the publications and citations associated with the search variable TS = (SMART CITY), an extensive body of literature was discovered, consisting of over thirty thousand publications that delve into and study the concept of Smart Cities.

As mentioned, Smart Tourism closely relates to the concept of a Smart City. To study the phenomenon of Smart Tourism, research was conducted on the publications and citations related to the search variable TS = (SMART TOURISM) as shown in Figure 3.

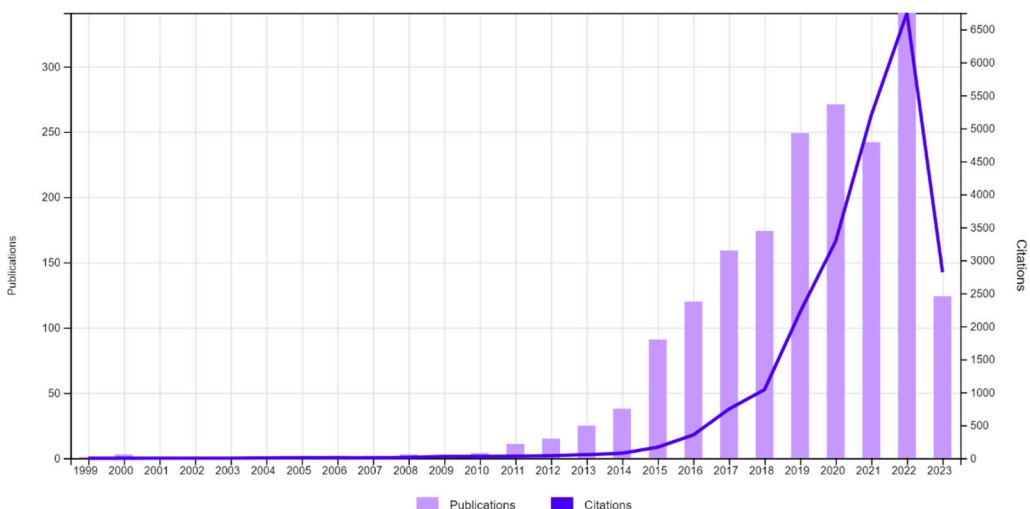

**Figure 3.** Studies related to Smart Tourism.

To accomplish this objective, research was performed in the Web of Science Core Collection, employing specific criteria to select journal articles, and considering the entire available period for analysis.

Research was conducted on both interconnected concepts, Smart Cities and Smart Tourism, and the following results were obtained. Figure 4 shows an overview of the existent literature associated with the search variables TS = (SMART TOURISM) AND TS = (SMART CITIES). The primary objective is to present the scientific publications relevant to these concepts. To accomplish this objective, research was performed in the Web of Science Core Collection, employing specific criteria to select journal articles and considering the entire available period for analysis.

### 2.3. Smart Cities, Smart Tourism, and Technology User Acceptance (TAM)

Having reviewed the literature on the concepts of Smart Tourism and Smart Cities, the research leads to evaluating the studies that connect these terms with the TAM.

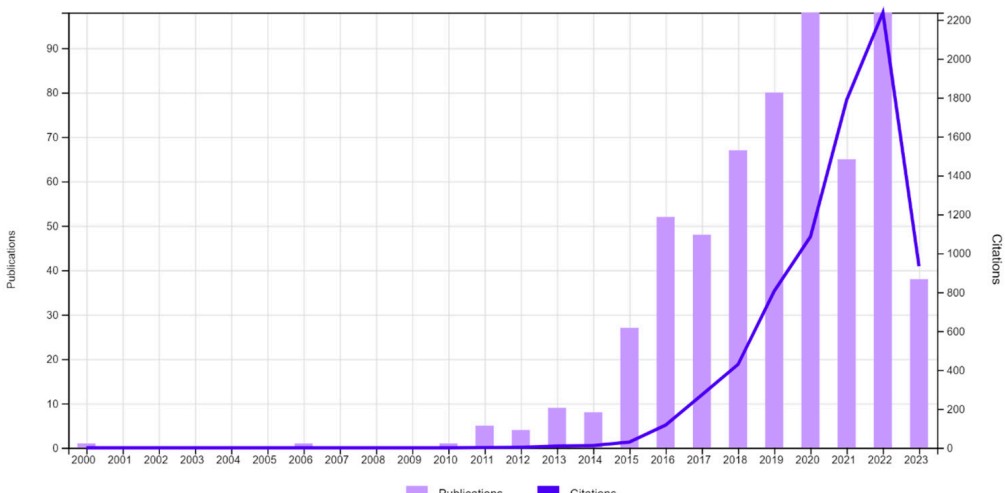

**Figure 4.** Studies related to Smart Cities and Smart Tourism.

Figure 5 provides a thorough summary of the research papers and references related to the search parameters, TS = (SMART TOURISM)) AND ((TS = (TECHNOLOGY ACCEPTANCE MODEL) OR TS = (TAM)). The main objective is to showcase the scientific publications relevant to this concept. To accomplish this objective, research was performed in the Web of Science Core Collection, employing specific criteria to select journal articles and considering the entire available period for analysis.

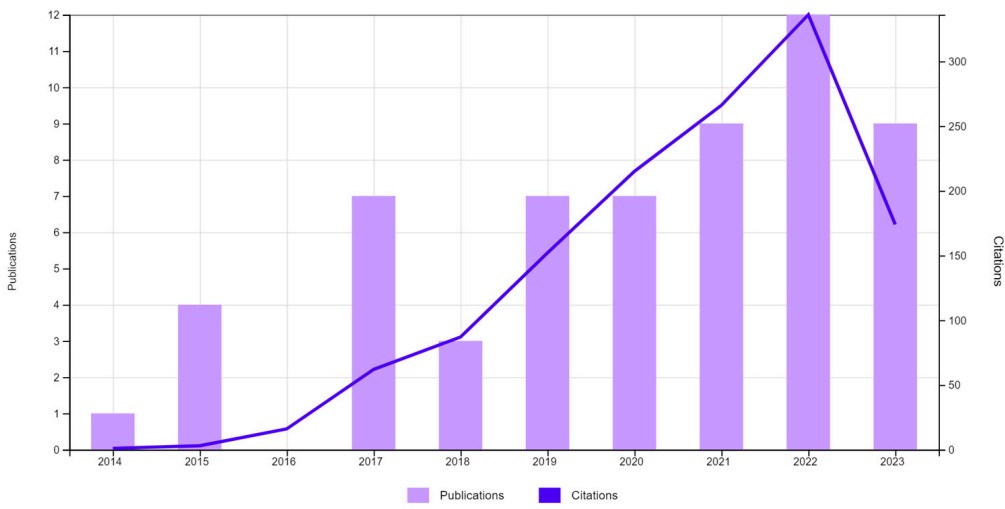

**Figure 5.** Studies related to Smart Tourism and TAM.

As observed in these sections, a substantial amount of literature exists on Smart Cities, Smart Tourism, and technology acceptance models. Nevertheless, upon reviewing the literature regarding the integration of these concepts, it becomes evident that there is a noticeable scarcity of scientific studies addressing this aspect as can be seen in Table 1.

**Table 1.** Studies related to Smart Cities, Smart Tourism and TAM.

| Year | Title |
|------|-------|
| 2018 | Framing a Smart Service with Living Lab Approach: A Case of Introducing Mobile Service within 4G for Smart Tourism in Taiwan [23]. |
| 2019 | The Augmented Reality in Lisbon Tourism Proposal for a AR Technology Adoption Model [24]. |

**Table 1.** *Cont.*

| Year | Title |
|------|-------|
| 2020 | The Role of Human–Machine Interactive Devices for Post-COVID-19 Innovative Sustainable Tourism in Ho Chi Minh City, Vietnam [25]. |
| 2022 | Using UTAUT-3 to Understand the Adoption of Mobile Augmented Reality in Tourism (MART) [26]. |
| 2023 | Small-Town Citizens' Technology Acceptance of Smart and Sustainable City Development [27]. |

Given the existing gap in the literature, a functional methodology is presented to develop a recommendation system for categorizing tourists based on their digital proficiency and their utilization of mobile technologies in Smart Cities.

### 2.4. Criteria for Measuring Technology User Acceptance

Numerous studies in the literature have identified various criteria for measuring technology user acceptance models. For this work and model proposal, the approach is based on the most frequently utilized TAM criteria, forming the foundation of the research:

- Frequency of Mobile Use (FMU): Some studies suggest that the frequency of mobile use is an important factor in technology acceptance models. This study [28], revealed that the perceived effectiveness and perceived convenience of mobile applications were impacted by crucial security elements. In this other study [29], the research discovered that both the perceived simplicity of usage and the portability aspects had a considerable impact on the perceived usefulness of mobile health services. In [30], it was revealed that the perceived level of physical risk and key factors from the TAM were strong indicators of individuals' intentions to use mobile applications for online transportation services. Finally, in [30], it was demonstrated that mobile social media usage exerted a substantial indirect influence on online business models. This influence was mediated by the TAM, emphasizing its critical role in the frequency of mobile use.

- Mobile App Usage (MAU): After a deep analysis of the existent literature, it can be highlighted that the impact of the mobile apps usage has directly influenced the TAM and that mobile app usage has had a positive impact on teacher performance and learning capabilities [31]. In this other study [32], it was observed that the perceived ease of use and perceived usefulness of mobile apps positively affected hotel consumers' experiences. Additionally, this study highlighted that perceived usefulness, along with user experience, played a significant role in influencing customers' acceptance of hotel apps. The papers suggest that TAM can be used to investigate the impact of mobile app usage in various contexts, such as education, hospitality, and conferences.

- Digital Competence (DC): The literature suggests that digital competence directly affects technology acceptance models. This study [33], incorporates technology readiness into the TAM and identifies that the influence of technology readiness on use intention is mediated by the perceptions of usefulness and ease of use. In other words, the impact of an individual's technological readiness on their intention to use a particular technology is influenced by how they perceive its usefulness and ease of use. In the same vein [34], extends the TAM by incorporating two types of perceived usefulness and reveals that perceived near-term usefulness has the most significant influence on the behavioral intention to use a technology. However, it is worth noting that perceived long-term usefulness also exerts a positive impact on the intention to use the technology, albeit to a lesser extent.

- Attitude towards Technology (ATT): This criterion is an important factor in technology acceptance models, its significance lies in gauging the emotional and cognitive responses of users, capturing their perceptions, beliefs, and attitudes towards technology. In [35], it was found that awareness and perceived risk are external variables that

affect the technology acceptance model for mobile banking in Yemen. On another hand, in this study [36], the research indicated that attitude served as a significant predictor of university students' intention to use e-learning, drawing from the TAM;

- Perceived Usefulness (PU) and Perceived Ease of Use (PEU): These two variables have been deeply considered and analyzed by authors who collectively suggest that perceived usefulness is an important factor in the TAM. In [37], it was discovered that perceived usability holds crucial importance in the TAM and that its presence explains a greater amount of variance in the model compared to its absence. In [38], it was found that perceived ease-of-use (PEU) is strongly related to perceived usability, which is a component of the modified TAM (mTAM). However, Ref. [39] proposed a theoretical model that suggests perceived ease of use is determined by control, intrinsic motivation, and emotion, and that it adjusts over time to reflect objective usability and perceptions of external control. This suggests that perceived usability may be more complex than simply incorporating it into the TAM. In this arena, many works suggest that perceived usefulness is a key factor in the TAM, but the relationship between perceived usability and TAM may require further investigation.
- Social Media Usage (SMU): The literature suggest that the TAM can be used to understand social media usage behavior. In this paper [40], interesting results highlighting the relationship between TAM and social media usage were found. The author found that the individual adoption behavior of Facebook can be explained by its perceived ease of use, critical mass, and social networking site capability. In a study conducted by Al-Qaysi [41], the author conducted a thorough systematic review, analyzing 57 research articles. The findings indicated that several factors significantly extended the TAM, among the most frequent factors were social media, perceived enjoyment, subjective norm, self-efficacy, perceived critical mass, perceived connectedness, perceived security, and perceived trust. In the same vein, in this other study [26], the digital gap that might exist across different generations was delved into and it was uncovered that age plays a significant role in influencing the optimism, innovativeness, and perceived usefulness concerning the adoption of social media. Finally, study [42] constructed a comprehensive model to investigate the influence of social-media-use factors on electronic banking adoption. They observed a notable negative impact of the social media factor on the expected efforts, which, in turn, affected the use of electronic banking services.
- Previous Experience (PE): The last criterion identified in this study is previous experience. In study [43], a meta-analysis of TAM was conducted which found that subjective norm (such as experience) has a significant influence on perceived usefulness and the behavioral intention to use any technology. On the other hand, in study [44], out a literature review of the technology acceptance models was carried out, and it was evident that these models play a vital role in comprehending the predictors of human behavior concerning the potential adoption or rejection of innovations and technologies. Finally, and in the same arena, in study [45], a research plan dedicated to exploring prospective interventions both before and after IT implementation was outlined, aiming to improve employees' acceptance and utilization of technology. Moreover, the studies suggest that prior experiences could influence technology acceptance models, and leveraging this criterion can foster technology adoption and utilization.

In this section, the proposed methodology for creating a novel model is introduced, representing a significant milestone in the existing literature.

## 3. Methodology

The objective of this section is to present a comprehensive theoretical framework for the research. To achieve this, the study will incorporate the following models: the analytic hierarchy process (AHP) and the 2-tuple fuzzy linguistic model (LD2T). It will also incorporate strategies to effectively manage heterogeneous information in the decision-

making process [46]. This integration of models will serve as the foundation for the proposed research, facilitating a deeper understanding and systematic analysis of the subject matter.

In certain scenarios, the data necessary for decision making originates from diverse expression domains. In the specific case under investigation, the expression domains to be utilized encompass both numerical and categorical data. Therefore, it becomes vital to tackle the problem by employing a fuzzy linguistic domain, which enables the unification of information management in the proposed model. The utilization of a fuzzy linguistic approach allows for a more flexible and comprehensive handling of heterogeneous data, leading to more effective and accurate decision-making outcomes [47].

### 3.1. 2-Tuple Model (LD2T)

The 2-tuple linguistic model (LD2T) is widely recognized as a prominent approach for computing with words [48]. In traditional decision-making models, information is typically represented by crisp numbers or linguistic terms, which often fail to capture the uncertainties and ambiguities associated with human perception and language. However, the LD2T addresses this limitation by considering both the linguistic term and its associated membership function.

By combining linguistic terms and fuzzy sets, the LD2T provides a more accurate representation of information [49]. It allows decision makers to express their preferences and perceptions using linguistic terms while incorporating the inherent fuzziness and vagueness of human language. The membership function associated with each linguistic term captures the degree of membership or relevance of the term to a specific decision or context.

This approach not only improves the accuracy of information representation but also enables more effective decision making. Decision makers can evaluate and compare alternatives based on a comprehensive understanding of the uncertainties and preferences involved. The LD2T has been successfully applied in various domains, including risk assessment, project management, and supplier selection, demonstrating its effectiveness in capturing complex and uncertain information.

Overall, the LD2T offers a promising framework for decision making, providing a more precise and realistic representation of information and supporting informed and robust decision-making processes [50].

The primary objective of this model is to improve the precision of information representation by compressing it into linguistic values $(s_i, \alpha_i)$, where $s_i \in S$ and $\alpha_i \in [-0.5, 0.5)$. The 2-tuple linguistic representation model commonly employs a triangular function to represent the selected membership function [50]. A visual depiction of the domain within s5, utilizing this triangular function, can be found in Figure 6 as an illustrative representation.

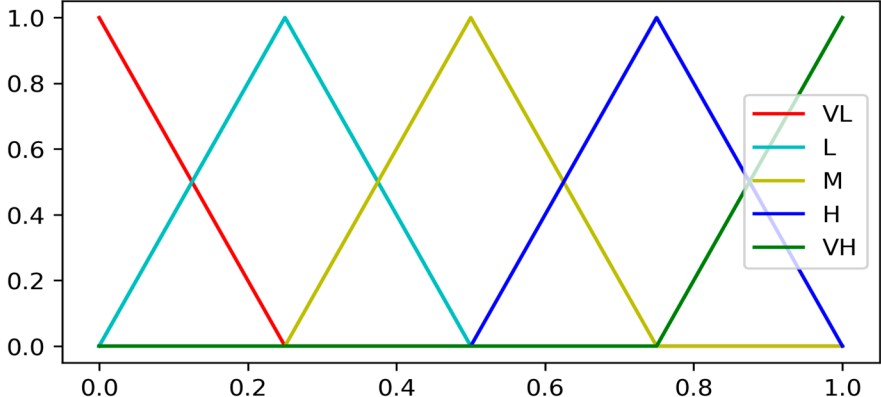

**Figure 6.** Triangular linguistic representation in an s5 domain.

**Definition 1.** *The linguistic domain in a 2-tuple linguistic model is a set denoted as S, which comprises linguistic terms used to represent information. In this context, the symbolic representation of a linguistic item, denoted as $s_i$, is a numerical value falling within the interval $[-0.5, 0.5)$. This value represents the difference between a given set of information, which is indicated by the numerical value $\beta \in [0, k]$, obtained through a calculation process, and its closest integer value, denoted as $i \in \{0, \ldots, k\}$.*

**Definition 2.** *Let $S = \{s_0, \ldots, s_l\}$ a set of linguistic terms, and $\langle S \rangle = S \times [-0.5, 0.5)$, and $\beta \in [0, l]$ that represents the result of an operation; the linguistic 2-tuple expressing the information equivalent to $\beta$ is obtained using the expression:*

$$\Delta_S : [0, l] \to \langle S \rangle$$
$$\Delta_S(\beta) = (s_i, \alpha_i), \quad \begin{cases} i = round(\beta) \\ \alpha = \beta - i, \alpha \in [-0.5, 0.5), \end{cases} \tag{1}$$

*In the given context, where round$(\cdot)$ denotes standard rounding operation, $s_i$ represents the label closest to $\beta$, and $\alpha$ represents the outcome of the symbolic translation, the result is a 2-tuple within the set $\langle S \rangle$ that corresponds to a value within the interval $[0, l]$.*

**Definition 3.** *Let $S = \{s_0, \ldots, s_l\}$ be a collection of linguistic terms, and $(s_i, \alpha_i) \in \langle S \rangle = S \times [-0.5, 0.5)$ corresponds to the linguistic value expressed in a 2-tuple format; $(s_i, \alpha_i)$ can be obtained by employing the following expression:*

$$\Delta_S^{-1} : \langle S \rangle \to [0, l]$$
$$\Delta_S^{-1}(s_i, \alpha_i) = i + \alpha = \beta \tag{2}$$

*The computational model associated with this analysis can be thoroughly examined. The following operators are defined, which help to understand and apply the model.*

Comparison operators: Given two 2-tuple linguistic values $(s_n, \alpha_1)$ and $(s_k, \alpha_2)$:
- If $n < k$, then $(s_n, \alpha_1)$ is less than $(s_k, \alpha_2)$;
- If $n = k$, then

  (a) If $\alpha_1 = \alpha_2$, then $(s_n, \alpha_1)$ and $(s_k, \alpha_2)$ symbolize identical information.
  (b) If $\alpha_1 < \alpha_2$, then $(s_n, \alpha_1)$ has a lower value than $(s_k, \alpha_2)$.
  (c) If $\alpha_1 > \alpha_2$, then $(s_n, \alpha_1)$ is larger than $(s_k, \alpha_2)$.

Aggregation operators: Let $S$ be a collection of 2-tuple linguistic values in $\langle S \rangle$, and let $\omega = (\omega_1, \ldots, \omega_k)$ represent their respective weights, such that the sum of the weights $\sum_1^k \omega_i = 1$. In this scenario, the weighted average of the 2-tuples is defined as $F^\omega \langle S \rangle^k : \to \langle S \rangle$.

$$F^\omega((s_1, \alpha_1), \ldots, (s_k, \alpha_k)) = \Delta_S \left( \sum_1^k \omega_i \Delta_S^{-1}(s_i, \alpha_i) \right) \tag{3}$$

*3.2. AHP Method*

The method known as the analytic hierarchy process (AHP) was devised by Thomas Saaty and serves as a decision-making technique [51]. It is used to solve complex and multi-criteria problems by hierarchically structuring the factors involved in the decision and systematically comparing the available options. AHP is applicable to a wide range of decision-making contexts, including business, engineering, healthcare, and environmental planning.

One aspect to consider in applying AHP is the number of experts involved in the decision-making process. In certain cases, decisions can be made by a single expert, while in others, a group of experts may be involved. When multiple experts are engaged, the AHP process incorporates their diverse perspectives and knowledge, leading to a more comprehensive and robust decision outcome.

Another factor influencing the application of AHP is the decision environment. Decision-making problems can be classified into structured and unstructured environments. In structured environments, the decision criteria and their relationships are well-defined and can be quantitatively evaluated. AHP is particularly effective in structured environments as it enables the systematic comparison and weighting of criteria, facilitating decision making.

On the other hand, unstructured decision environments involve subjective criteria and uncertain relationships between them. The AHP can still be applied in these situations, but additional considerations such as expert judgment and qualitative assessments play a significant role in determining the relative importance of the criteria and alternatives.

Additionally, the number of criteria in a decision problem affects the complexity of the AHP analysis. The problems with a small number of criteria are relatively simpler to handle, allowing for straightforward pairwise comparisons and the derivation of priority weights. As the number of criteria increase, the pairwise comparisons become more numerous and intricate, demanding additional effort in the analysis.

To address the challenges posed by large-scale problems, techniques such as hierarchical structuring and decomposition can be employed within the AHP framework. These techniques break down complex decision-making problems into manageable subproblems, facilitating the decision-making process.

The AHP method offers a versatile approach to decision making that can be adapted to various scenarios, and has been specially developed to address intricate decision scenarios involving multiple criteria [52]. The involvement of multiple experts, consideration of the decision environment, and the number of criteria are key factors in utilizing the AHP effectively. Understanding these aspects allows decision makers to classify decision problems and apply the appropriate techniques to achieve informed and rational decisions.

In this study, the AHP is chosen because it is a suitable methodology for evaluating and ranking different criteria and options in the context of recommender systems and Smart Tourism. The AHP is particularly useful when criteria are not easily quantifiable or when judgments are subjective, which is common in situations related to tourism and user preferences.

Its compatibility with the fuzzy linguistic approach lies in the fact that both approaches can handle uncertainty and an ambiguity of information. The AHP can handle numerical values obtained from subjective judgments, while the fuzzy linguistic approach uses linguistic terms to express uncertainty. By combining both methodologies, a more comprehensive and accurate evaluation can be achieved, as the fuzzy linguistic approach complements the limitations of the AHP by capturing preferences and uncertainties more expressively and richly in linguistic terms [53]. This allows for a more realistic and suitable representation of data in situations where information is vague or ambiguous, such as in the context of tourism and user preferences in Smart Cities.

AHP provides the decision-making framework, while LD2T enhances it by accommodating heterogeneous and uncertain data, making their combination valuable for handling complex and diverse decision-making scenarios in Smart Cities and tourism contexts.

Within the framework of the suggested model, the assessment of each criterion's weightage will play a crucial role in determining the level of digital maturity of tourists in a Smart City setting.

Further details and elaboration on the entire process are provided in the following subsections.

### 3.2.1. Organizing the Decision Model in a Hierarchical Process

The AHP approach initiates by organizing the decision problem into a hierarchical structure, facilitating an effective problem analysis and organization. This hierarchical representation categorizes the problem into different levels based on shared attributes, allowing for a thorough assessment of decision factors.

At the highest level of the hierarchy, the objective or target is situated, representing the primary goal that guides the decision-making process.

The second level comprises a set of criteria, $C = \{c_1, \ldots, c_{\#c}\}$. These criteria serve as the fundamental factors or dimensions for consideration in the decision. Each criterion can be further subdivided recursively into sub-criteria, $c_{ij}$, $c_{1j} = \{c_{11}, \ldots, c_{1\#C1}\}$.

Finally, the lowest level of the hierarchy comprises alternatives or choices available for consideration in the decision-making process, $A = \{a_1, \ldots, a_{1\#A}\}$.

Figure 7 presents a visual representation of the hierarchical structure of the decision problem, showcasing the connections and dependencies among the objective, criteria, sub-criteria, and alternatives. This graphical illustration provides a clear and concise overview of the relationships among these elements, aiding in the decision-making process.

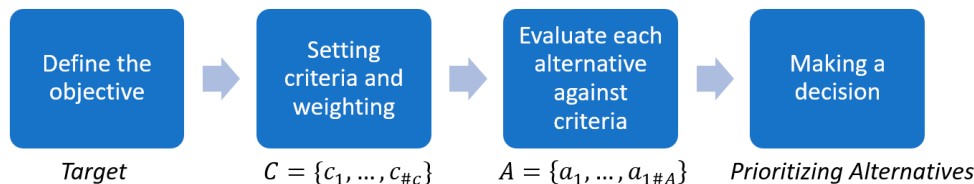

**Figure 7.** Hierarchical structure of the decision problem.

### 3.2.2. Setting Criteria and Weighting

In the AHP model, criteria and their corresponding weights can be established by utilizing the pairwise comparison matrix (*PW*), Saaty's scale of values (Table 2), weight vector, and consistency ratio (CR).

**Table 2.** Saaty's scale [54].

| Degree of Importance | Definition |
|---|---|
| 1 | Criteria of equal importance |
| 3 | Criteria with a moderate preference over another |
| 5 | Criteria with a substantial or strong preference over another |
| 7 | Criteria with a very strong or demonstrated preference over another |
| 9 | Criteria with an extreme preference over another |
| 2, 4, 6, 8 | Intermediate values used to express the preference of importance between criteria |
| Reciprocals | Inverse values |

To begin, the pairwise comparison matrix (*PW*), is constructed to compare each criterion against every other criterion, where the experts assign values to represent the relative importance or preference between criteria based on Saaty's scale, which ranges from 1 (indicating equal importance) to 9 (indicating extreme importance).

Next, the matrix is normalized by calculating the row geometric means or column sums to obtain the weight vector. The weight vector represents the relative importance of each criterion and serves as a basis for the decision making.

To ensure the consistency and reliability of the weights assigned, the consistency ratio is computed (CR). This ratio assesses the consistency of the pairwise comparisons made by experts. If the consistency ratio exceeds a predefined threshold, adjustments may be required to achieve more consistent judgments.

By considering the pairwise comparison matrix, Saaty's scale of values, the weight vector, and the consistency ratio, the criteria and their corresponding weights can be established in the AHP model. This systematic approach enhances the decision-making process by incorporating expert opinions and quantifying the relative importance of the criteria.

The vector representing the weights of criteria, denoted as $w$, is calculated using the eigenvector method which involves the following:

$$\sum_{j=1}^{n} pw_{ij}\omega_j = \lambda_{max} \times \omega_i \tag{4}$$

The maximum eigenvalue of the pairwise comparison matrix $PW$ is denoted as $\lambda_{max}$ and the corresponding normalized eigenvector linked to the principal eigenvalue of $PW$ is represented by $w$. These values play a crucial role in determining the weights of criteria in the decision-making process. The consistency ratio (CR) is a metric used to assess the reliability of the AHP method by evaluating the consistency of pairwise comparisons conducted by the decision maker throughout the AHP process. It helps ensure that the decision maker's judgments are consistent and reliable, contributing to the overall validity of the decision-making outcomes:

$$CR = CI/RI \tag{5}$$

The consistency ratio (CR) is determined by dividing the consistency index (CI), which is calculated as $\frac{\lambda_{max}-n}{n-1}$, by the random consistency index (RI) as listed in Table 3.

**Table 3.** Random consistency indices [54].

| $n$ | 1 | 2 | 3 | 4 | 5 | 6 | 7 | 8 | 9 | 10 |
|---|---|---|---|---|---|---|---|---|---|---|
| Random Consistency Index (RI) | 0.00 | 0.00 | 0.58 | 0.9 | 1.12 | 1.24 | 1.32 | 1.41 | 1.45 | 1.49 |

When the calculated consistency ratio (CR) meets or falls below the predefined consistency limits specified in Table 4, it indicates that the hierarchical comparisons' results satisfy the consistency criteria. In these situations, the pairwise comparison values are considered acceptable, ensuring a consistent decision-making process.

**Table 4.** Consistency limits [55].

| Size of the Consistency Matrix | Consistency Ratio |
|---|---|
| 3 | 5% |
| 4 | 9% |
| $\geq 5$ | 10% |

### 3.2.3. Evaluating Each Alternative against the Established Criteria

Similarly, employing a recursive approach, the same methodology can be utilized to assess the alternatives by establishing their relationships with each criterion [54]. This process allows us to determine the weights and preferences of the alternatives, enabling a comprehensive evaluation and comparison to support the decision-making process. This involves constructing a comparison matrix for the alternatives, denoted as $W$, with dimensions $n \times n$, in this context, each element denotes the weight assigned to one alternative concerning another alternative for a specific criterion.

Following the outlined methodology, the weights of the alternatives can be determined by analyzing the pairwise comparisons within each criterion. This involves finding the maximum eigenvalue ($\lambda_{max}$) and its corresponding eigenvector ($w$) for each comparison matrix ($W$). The eigenvector provides the relative weights of the alternatives for the specific criterion.

By iteratively applying this process for each criterion, it is possible to construct a matrix of weights that incorporates the comparisons of each alternative against all others, while taking all the criteria into account. This matrix of weights facilitates the ranking of alternatives based on the assigned weightings.

As previously described, the subsequent stages of the process involve using the acquired weights to calculate the overall contributions of each alternative in achieving the main objective. Consequently, a ranking of the alternatives can be derived based on their assigned weights.

This methodology guarantees a systematic and thorough evaluation of the alternatives, encompassing various criteria and their respective weightings. As a result, decision makers can prioritize the alternatives based on their relative significance and make informed decisions.

### 3.2.4. Decision Making

In the final stage of the decision-making process, the weights assigned to the alternatives, derived from expert evaluations of the criteria, are used to determine the overall ranking. This ranking is essential in identifying the most appropriate alternative that aligns with the research objective. By integrating expert assessments and criteria weightings, decision makers can make well-informed choices, selecting the alternative that best meets the desired objectives and criteria.

The limitations of a decision-making model involving different experts using AHP may include inconsistency in decisions, individual biases, lack of specific knowledge, and difficulty in achieving consensus.

However, the decision-making model involving different experts using AHP can also offer flexibility and a group-focused approach. By allowing the incorporation of weights for each decision maker, the model considers the relative importance of the opinions from different experts in the outcome. This can improve the acceptance and commitment from those involved in the decision-making process, leading to more balanced decisions backed by a group consensus.

### 3.3. Treatment of Heterogeneous Information

In this research project, the main aim is to bring together different types of information from diverse sources using a specialized linguistic information domain called the 2-tuple linguistic information domain [50]. To initiate this process, a fundamental set of linguistic terms, known as the basic set of linguistic terms (CBTL), is established, which serves as the groundwork for all subsequent analyses and computations in this study.

The identification and selection of the CBTL domain, represented as $\bar{S} = \{s_0, \ldots, s_k\}$, involve carefully determining the set of linguistic terms that offer the highest level of granularity within the diverse information framework [55]. Through this careful approach, the maximum amount of information encapsulated within the linguistic domain is preserved. Once the CBTL set is defined, the next step involves transforming the various expression domains to align with the chosen CBTL set. It is essential to recognize that information can be expressed in various domains, such as numerical, interval, and linguistic, as depicted in Figure 8.

By adopting this approach, the main objective is to construct a comprehensive and coherent representation of heterogeneous information and the primary aim is to achieve a comprehensive and coherent representation of heterogeneous information. Integrating different expression domains with the chosen CBTL set will enable the seamless integration and comparison of data from various sources. This data harmonization will result in more precise and meaningful insights, facilitating a deeper understanding of the underlying patterns and relationships within the information. Ultimately, this unified representation will enhance the ability to make informed decisions based on a holistic view of the heterogeneous data.

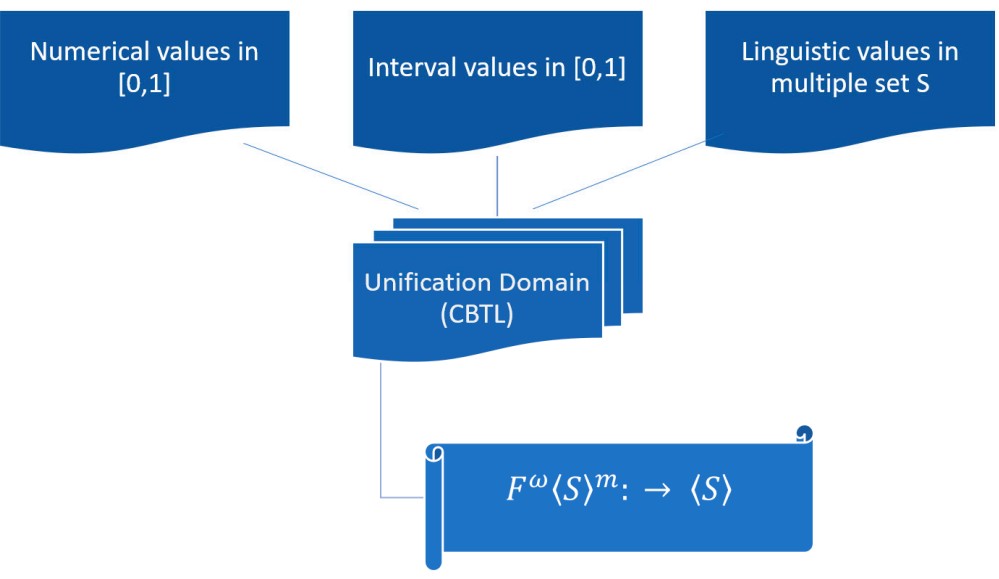

**Figure 8.** The 2-tuple linguistic unification model [50].

### 3.3.1. Numerical Domain

**Definition 4.** *Given a numerical value $n \in [0,1]$ and a set $\bar{S} = \{\bar{s}_0, \ldots, \bar{S}_g\}$ belonging to the CBTL domain, the numerical transformation function is defined, $T_{N\bar{S}} : [0,1] \rightarrow F(\bar{S})$:*

$$T_{N\bar{S}}(n) = \left\{ (\bar{S}_0, \omega_0), \ldots, (\bar{S}_g, \omega_g) \right\}, \ \bar{S}_i \in \bar{S} \text{ and } \omega_i \in [0,1] \tag{6}$$

*with*

$$\gamma_i = \mu_{\bar{S}_i}(n) = \begin{cases} 0, & if \ n \notin Support\left(\mu_{\bar{S}_i}(x)\right) \\ \frac{n - a_i}{b_i - a_i}, & if \ a_i \leq n \leq b_i \\ 1, & if \ b_i \leq n \leq d_i \\ \frac{c_i - n}{c_i - d_i}, & if \ d_i \leq n \leq c_i \end{cases} \tag{7}$$

*where $\gamma_i = \mu_{\bar{S}_i}(n) \in [0,1]$ is the degree of association between the numerical value n and $\bar{S}_i \in \bar{S}$.*

### 3.3.2. Interval Domain

**Definition 5.** *Considering a given value $u = [a, b] \in P([0,1])$ and a set $\bar{S} = \{\bar{S}_0, \ldots, \bar{S}_k\}$ belonging to the CBTL domain, the interval transformation function is defined $T_{I\bar{S}} : P([0,1]) \rightarrow F(\bar{S})$:*

$$T_{I\bar{S}}(u) = \left\{ \left( \bar{S}_j, \gamma_j^i \right) / k \in \{0, \ldots, k\} \right\} \tag{8}$$

*where $\gamma_k^i = max_y min\left\{ \mu_I(y), \ \mu_{\bar{S}_k}(y) \right\}$ and $\mu_I(y)$ and $\mu_{\bar{S}_k}(y)$ represent the membership functions corresponding to the interval I and the linguistic terms $\bar{S}_k$, respectively.*

$$\mu_I(y) = \begin{cases} 0 \ if \ y < a \\ 1 \ if \ a \leq y \leq b; \ y \in [0,1] \\ 0 \ if \ y > b \end{cases} \tag{9}$$

### 3.3.3. Linguistic Domain

**Definition 6.** *Considering the set $S = \{l_0, \ldots, l_m\}$ and the set $\bar{S} = \{\bar{S}_0, \ldots, \bar{S}_n\}$, belonging to the CBTL domain $\bar{S} = \{\bar{S}_0, \ldots, \bar{S}_n\}$, where both sets represent linguistic terms, and $n \geq m$, the function responsible for linguistic transformation, $T_{S\bar{S}} : S \rightarrow F(\bar{S})$ is defined as:*

$$T_{S\bar{S}}(l_i) = \left\{ \left( \bar{S}_k, \gamma_k^i \right) / k \in \{0, \ldots, n\} \right\} \ \forall \, l_i \in S \tag{10}$$

*where $\gamma_k^i = max_y min \left\{ \mu_{l_i}(y), \ \mu_{\bar{S}_k}(y) \right\}$, $i = 0, \ldots, n$ and $\mu_{l_i}(y)$ and $\mu_{\bar{S}_k}(y)$ identify the membership functions corresponding to each term $l_i$, $\bar{S}_k$.*

After consolidating the diverse information into a 2-tuple linguistic domain, a variety of specific operations become possible within the LD2T domain. These operations lead to interpretable outcomes that merge the diverse evaluations within a unified domain, specific to each criterion.

## 4. Proposed Model

In this study, a set of criteria was established, guided by the relevant literature, that directly measure the digital maturity level of the tourist (DMT) and facilitate the customization of the Customer Journey in Smart Cities.

The fuzzy linguistic model, LD2T, is necessary to handle heterogeneous data in the specific context of Smart Tourism and Smart Cities due to the inherent uncertainty and vagueness present in decision-making processes within these domains. In the context of Smart Tourism and Smart Cities, decision making involves dealing with a diverse range of preferences and subjective judgments from citizens and tourists. Additionally, the data obtained from different sources can be imprecise and difficult to quantify, making it challenging to apply traditional quantitative models effectively.

LD2T is well-suited for handling such heterogeneous data because it allows decision makers to express information in linguistic terms, capturing imprecision and uncertainty more effectively. Instead of relying solely on precise numerical values, LD2T uses linguistic variables to represent the subjective evaluations, preferences, and perceptions of users and citizens. These linguistic terms are accompanied by degrees of membership that represent the level of certainty or confidence in the linguistic expressions.

The combination of AHP and LD2T in the context of Smart Tourism and Smart Cities enhances the decision-making process. AHP provides a structured framework for the hierarchical analysis and pairwise comparisons of criteria and alternatives, while LD2T enriches this process by capturing heterogeneous data in the form of linguistic variables and degrees of membership.

In practical scenarios, when evaluating Smart Tourism initiatives, AHP and LD2T can help prioritize projects that align with citizens' preferences and values. For example, in a Smart City context, if the goal is to improve tourist experiences, AHP can analyze criteria such as technology accessibility, sustainability, and safety. LD2T can then be employed to capture tourists' linguistic preferences and perceptions about these criteria, allowing decision makers to tailor personalized interactions and services based on the fuzzy linguistic evaluation.

The complete process of determining the maturity level, based on a set of simulated data, is illustrated in Figure 9.

The new digital maturity level of tourist (DMT) measurement model proposed in this document is based on the following criteria (selected in Section 2):

- FMU (mobile device usage frequency): this variable represents the frequency with which tourists use mobile devices (such as smartphones or tablets) in their daily lives.
- MAU (usage of tourism mobile applications): this variable reflects tourists' willingness to use specific mobile applications to access tourism information, make reservations, obtain recommendations, etc.

- DC (competence in technology use): these variable measures tourists' competence and ability to use digital technologies in general.
- ATT (attitude towards technology adoption): this variable reflects tourists' attitude towards adopting technology in the tourism context.
- SMU (social media usage and content sharing): this variable represents the level of tourists' engagement in social media and the frequency with which they share content related to their tourism experiences.
- PE (previous experience with tourism technology): this variable indicates whether tourists have previous experience in using tourism technology, such as mobile applications, online bookings, digital travel guides, etc.
- PU (perception of usefulness): this variable reflects the tourist's perception of the usefulness of technology in the tourism context, how technology enhances their tourism experience, provides useful information, and meets their needs and expectations.
- PEU (perception of ease of use): this variable represents the tourist's perception of the ease of use of technology, how easy it is for them to use technology, navigate through mobile applications, access information, and perform actions.

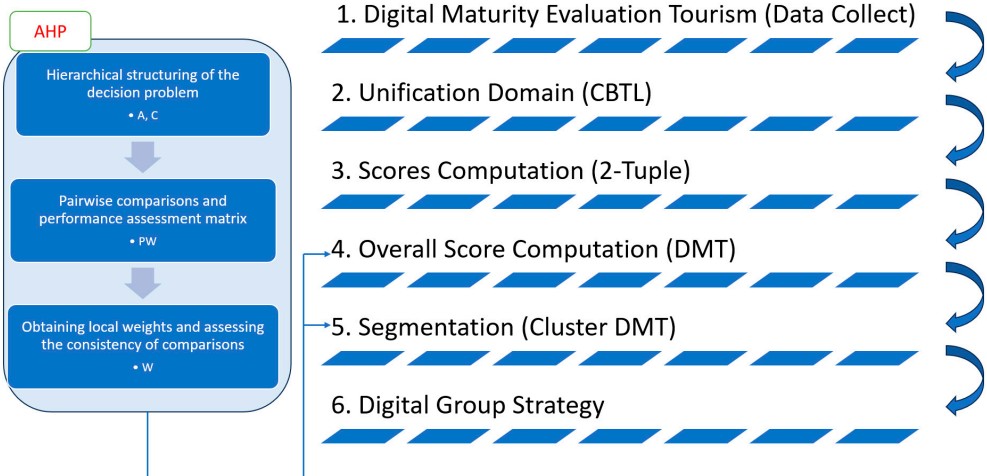

**Figure 9.** DMT model.

The identified criteria are utilized to evaluate the digital maturity level of tourists. This assessment provides valuable information that contributes to the development of personalized and real-time Customer Journeys.

Therefore, the first of the assessments of the tourist's level of digital maturity is represented in terms of the parameters described: mobile device usage frequency (FMU), usage of tourism mobile applications (MAU), competence in technology use (DC), attitude towards technology adoption (ATT), social media usage and content sharing (SMU), previous experience with tourism technology (PE), perception of usefulness (PU), and perception of ease of use (PEU). This is so that the assessment (DMT) can be expressed as $DMT = f(FMU, MAU, DC, ATT, SMU, PE, PU, PEU)$.

All criteria defined in the model are derived from the thorough literature review conducted in Section 2 of this study. The process described in Figure 9, is as follows:

1. Data collection: For this step, the process entails collecting the pertinent data and information that are relevant to the variables or criteria being studied. All the variables that constitute the model are defined within a range of values from 0 to 4.
2. Determine the CBTL domain of expression for each criterion: This involves defining linguistic terms or categories that represent the different levels or degrees of each criterion. In this study, and considering the specific use case, a scale consisting of five values {*very low, low, moderate, high, very high*} will be employed. As this scale pertains to linguistic expressions, it will be modeled using the set S.

3. Scores computation 2-tuple: The 2-tuple model, which is a fuzzy logic-based approach, can then be applied to the collected data to handle linguistic uncertainty and quantify the degree of membership for each linguistic term. For each evaluation, the variable must be calculated, $FMU_i$, $MAU_i$, $DC_i$, $ATT_i$, $SMU_i$, $PE_i$, $PU_i$, $PEU_i \in S \times [-0.5, 0.5)$. To optimize the utilization of the computational model, this data domain will be converted into 2-tuple linguistic variables.

4. Obtain the global score for each interaction using the AHP model: The AHP model is utilized to calculate a global score for each tourist based on the weighted of the different criteria. During this stage, the value of the 2-tuple, $E_k$, that characterizes the score of each evaluation is calculated using Equation (3), so that $E_k = DMT_k = F^\omega[A_{ki}]$. Next, the decision problem will be set up using a hierarchical model (AHP). Subsequently, the pairwise comparison matrix will be constructed, which can be represented by Equation (4), to obtain the weights vector for each variable, $W = w_{FMU}, w_{MAU}, w_{DC}, w_{ATT}, w_{SMU}, w_{PE}, w_{PU}, w_{PEU}$. This approach helps to weigh the criteria for each of the variables that make up the model according to expert opinions.

5. Designate the clusters that identify the different levels of digital development: the interactions can be grouped into clusters based on their similarities and differences in terms of the identified levels of digital development.

6. Develop a customized Customer Journey process for each cluster: This involves designing tailored experiences, strategies, or interventions that cater to the specific characteristics, preferences, and needs of each cluster. This can help optimize the digital development and overall satisfaction of tourists within each cluster.

## 5. DMT Model, Practical Application

In this section, an illustrative example will be presented of the application of the new DMT (digital maturity tourism) model to four different segments of potential users, classified by age: 26–35 years, 36–45 years, 46–55 years and 56–75 years. The dataset used for this analysis comprises a total of 600 records collected anonymously from various tour operators in Spain. The objective is to develop a working method that allows for the classification of the tourist profile according to their level of digital maturity, and from this point onwards, the generation of personalized Customer Journeys. Using this dataset, the developed methodology can be effectively applied its applicability can be extended to a variety of Smart City environments. This approach allows for the emphasis of the variables described in the model, thus improving the digital experience of tourists and exploring the use of technology in the field of Smart Tourism.

### 5.1. Data Collection

The data collected to evaluate the performance of the proposed model is shown in Figures 10–12.

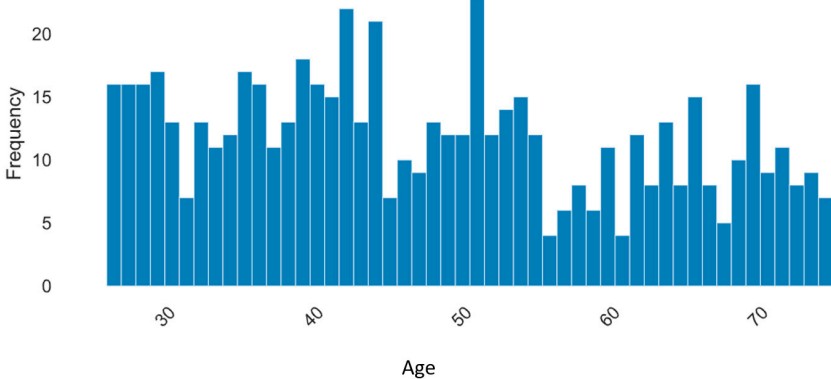

**Figure 10.** Frequency of age.

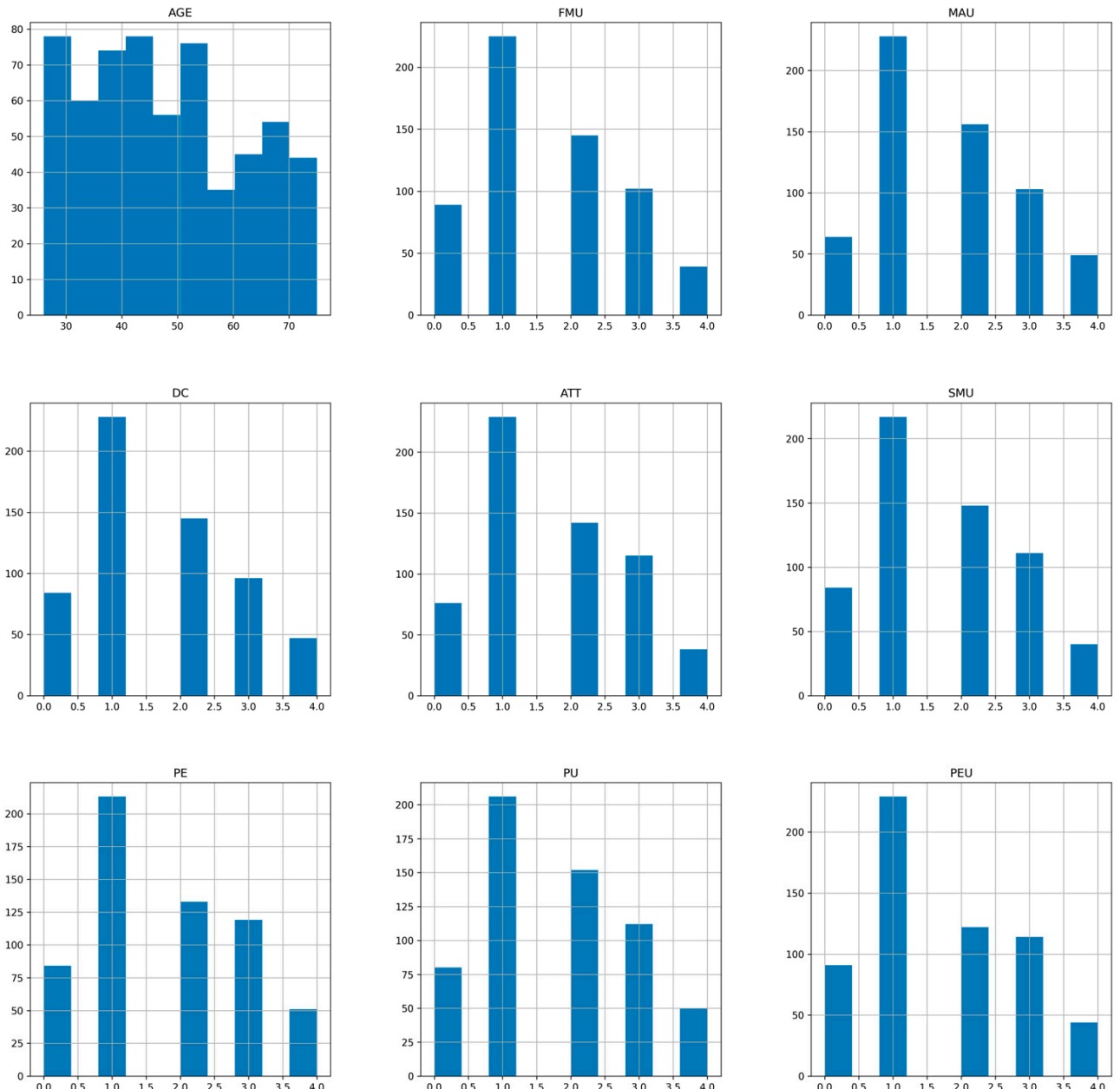

**Figure 11.** Histograms for AGE, FMU, MAU, DC, ATT, SMU, PE, PU, PEU.

It can be stated that all variables in the model exhibit a left-skewed or left-shifted Gaussian distribution. Figure 12 shows the distribution by age range of each of the variables that make up the model.

It can be observed that as age increases, the level of digitization and technology use among individuals tends to decrease. Consequently, the use of technology in Smart Cities is limited to certain age ranges. Through this exploratory data analysis (EDA), the need to establish different Customer Journeys linked to age groups in Smart Tourism becomes apparent.

### 5.2. CBTL Domain and Score Computation

The variables were grouped into a linguistic domain in Section 5, and the resulting outcomes are presented in Table 5. A representative set of data has been chosen with a total of 16 records. ID represents the identifier of each person who is part of the data sample.

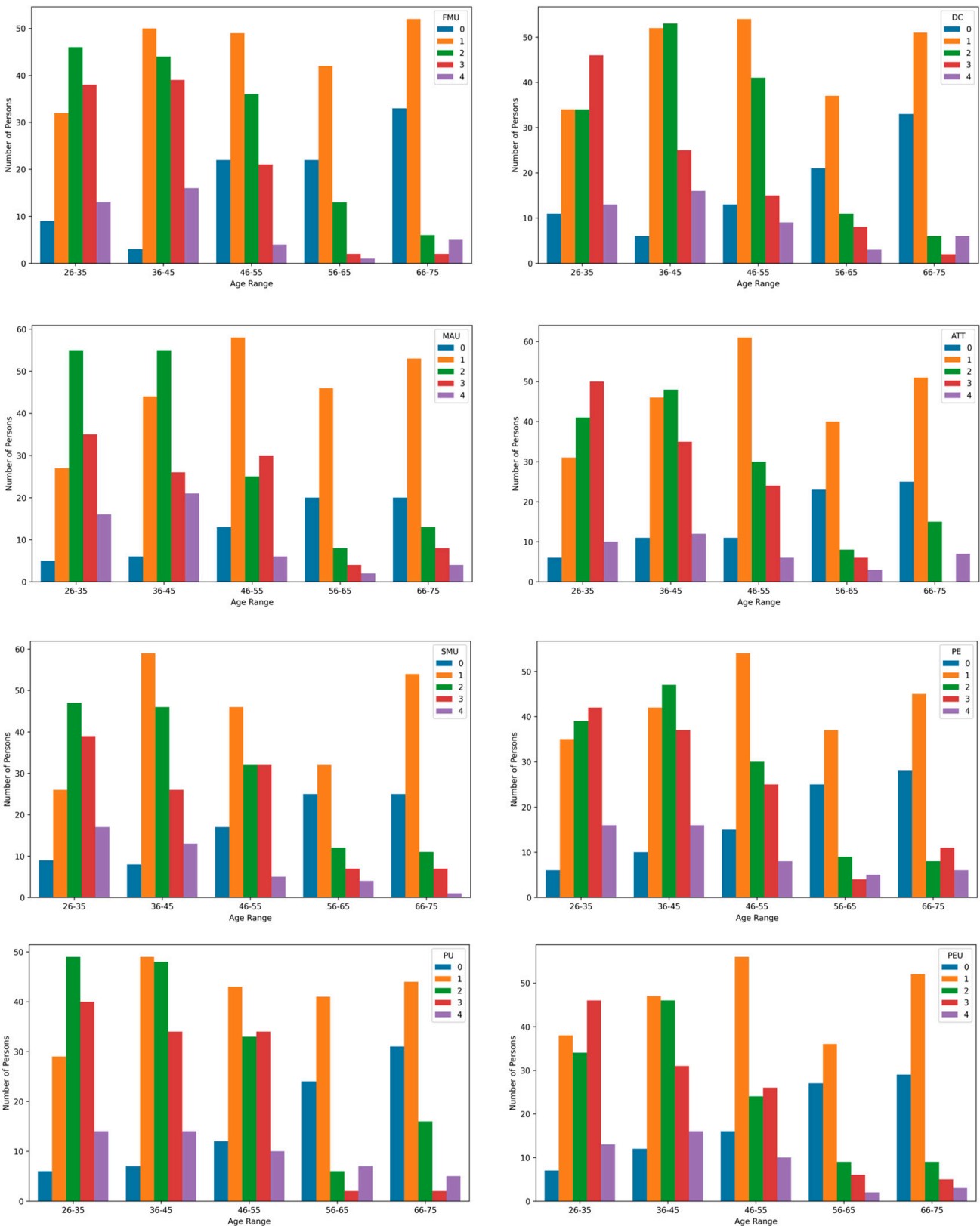

**Figure 12.** Bivariate analysis.

**Table 5.** DMT, matrix in the *S*5 domain.

| ID | AGE | FMU | MAU | DC | ATT | SMU | PE | PU | PEU |
|---|---|---|---|---|---|---|---|---|---|
| 0011 | 28 | M | M | VH | H | VH | L | VH | H |
| 0092 | 33 | M | H | VH | H | M | VH | VH | M |
| 0143 | 40 | H | M | VH | L | L | L | H | VL |
| 0147 | 42 | H | H | VH | H | VH | M | M | H |
| 0170 | 42 | M | M | M | L | M | M | H | M |
| 0198 | 41 | M | H | H | L | H | VH | H | VH |
| 0292 | 46 | H | L | M | M | VL | H | VL | VL |
| 0327 | 50 | H | L | L | L | VL | L | H | H |
| 0348 | 52 | VL | M | L | L | H | L | M | VH |
| 0439 | 60 | VL | VL | VL | L | L | VL | L | VL |
| 0451 | 62 | VL | L | L | L | VL | VH | L | L |
| 0496 | 64 | L | L | L | L | L | VL | L | L |
| 0513 | 62 | M | VL | L | L | L | VL | M | VL |
| 0529 | 66 | M | VL | L | L | VL | L | VL | VL |
| 0577 | 58 | L | L | L | VL | VL | L | H | VL |
| 0580 | 66 | M | M | L | VL | L | L | L | L |

### 5.3. DMT, Overall Score

At this stage of the study, it is essential to assess the relative significance of each feature of the DMT model before computing the overall interaction score. To achieve this, the AHP model will be utilized.

To establish a consensus, the group of consulted experts utilized the Saaty scale and created the following matrix (Table 2):

$$
W = \begin{bmatrix}
 & FMU & MAU & DC & ATT & SMU & PE & PU & PEU \\
FMU & 1 & 1 & 1/5 & 1/3 & 1/3 & 3 & 1/5 & 1/5 \\
MAU & 1 & 1 & 1/5 & 1/3 & 1/3 & 3 & 1/5 & 1/5 \\
DC & 5 & 5 & 1 & 3 & 3 & 5 & 1 & 1 \\
ATT & 3 & 3 & 1/3 & 1 & 3 & 3 & 1/3 & 1/3 \\
SMU & 3 & 3 & 1/3 & 1/3 & 1 & 1 & 1/5 & 1/5 \\
PE & 1/3 & 1/3 & 1/5 & 1/3 & 1 & 1 & 1/5 & 1/5 \\
PU & 5 & 5 & 1 & 3 & 5 & 5 & 1 & 1 \\
PEU & 5 & 5 & 1 & 3 & 5 & 5 & 1 & 1
\end{bmatrix}
$$

The individual hierarchical results produced satisfactory outcomes, and consistency was ensured when CR ≤ 0.10, as denoted in Equation (5). In this instance, CR equals 0.065, validating the accuracy of the model's results.

The final weightings obtained are as follows: $W = \{w_{FMU} = 0.049, w_{MAU} = 0.049, w_{DC} = 0.221, w_{ATT} = 0.108, w_{SMU} = 0.069, w_{PE} = 0.037, w_{PU} = 0.234, w_{PEU} = 0.234\}$.

Consequently, experts have assigned greater importance to perceived usefulness (PU) and perceived ease of use (PEU), followed by digital competence (DC), attitude towards technology (ATT), social media usage (SMU), frequency of mobile use (FMU), mobile app usage (MAU), and previous experience (PE). These variables, along with their respective weights obtained through the AHP model, contributed to the final score of the DMT model.

Table 6 presents the global score of the DMT model, set out in the previous steps.

The outlined procedure enables the derivation of individual scores for each person based on the designated criteria. This establishes a recommendation, prioritization, and personalization model that assesses the tourist's potential for utilizing technology in Smart City environments. It is important to emphasize that the methodology is not confined solely to the selected criteria but can be extended to encompass a broader range of criteria, sub-criteria, and diverse areas of numerical and linguistic representation. Furthermore, the approach can involve multiple decision makers in the decision-making process, providing a flexible and scalable framework to address a wide variety of complex

decision-making problems. As a result, the assessment obtained for each tourist can guide the prioritization of interactions, facilitate the effective utilization of technology within Smart Cities, and contribute to optimizing the various touchpoints along the tourist's customized Customer Journey.

**Table 6.** DMT, overall score.

| ID | AGE | FMU | MAU | DC | ATT | SMU | PE | PU | PEU | DMT |
|------|-----|-----|-----|----|-----|-----|----|----|-----|------------|
| 0011 | 28 | M | M | VH | H | VH | L | VH | H | (H, 0.089) |
| 0092 | 33 | M | H | VH | H | M | VH | VH | M | (H, 0.036) |
| 0143 | 40 | H | M | VH | L | L | L | H | VL | (M, 0.011) |
| 0147 | 42 | H | H | VH | H | VH | M | M | H | (H, 0.006) |
| 0170 | 42 | M | M | M | L | M | M | H | M | (M, 0.032) |
| 0198 | 41 | M | H | H | L | H | VH | H | VH | (H, 0.002) |
| 0292 | 46 | H | L | M | M | VL | H | VL | VL | (L, −0.009) |
| 0327 | 50 | H | L | L | L | VL | L | H | H | (M, −0.008) |
| 0348 | 52 | VL | M | L | L | H | L | M | VH | (M, 0.019) |
| 0439 | 60 | VL | VL | VL | L | L | VL | L | VL | (VL, 0.103) |
| 0451 | 62 | VL | L | L | L | VL | VH | L | L | (L, −0.002) |
| 0496 | 64 | L | L | L | L | L | VL | L | L | (L, −0.009) |
| 0513 | 62 | M | VL | L | L | L | VL | M | VL | (L, −0.009) |
| 0529 | 66 | M | VL | L | L | VL | L | VL | VL | (VL, 0.116) |
| 0577 | 58 | L | L | L | VL | VL | L | H | VL | (L, 0.015) |
| 0580 | 66 | M | M | L | VL | L | L | L | L | (L, −0.002) |

The criteria serve as a roadmap for tailoring the Customer Journey of tourists visiting Smart Cities.

### 5.4. DMT, Clustering

The variables in the extended DMT model are represented in a 2-tuple format, which facilitates straightforward interpretation and aggregation processes while retaining all relevant information. Traditional clustering methods typically operate within a numerical domain, and, in this case, the function $\Delta^{-1}$ will be utilized as defined in Equation (2). This function enables an effective analysis and clustering based on the given variables.

The correlation coefficient of −66 between age and the DMT scoring indicates a moderate correlation as shown Figure 13. This correlation aligns with expectations, as it is commonly observed that as individuals grow older, their ability to adapt to technology tends to decrease. Therefore, in the context of Smart Tourism, there is a greater need for personalized guidance and assistance for older individuals to navigate and fully benefit from technology-enabled experiences.

Although the use of an algorithm such as k-means in this case is not advisable, it helps to visually distinguish the three main groups as shown in Table 7, which can also be understood within the exploratory data analysis that was carried out.

**Table 7.** Cluster k-means algorithm represented in 2-tuple.

| Cluster c | AGE | DMT | Tourist |
|-----------|-------------|-------------|---------|
| 0 | (M, −0.023) | (M, −0.071) | 220 |
| 1 | (L, −0.084) | (M, 0.036) | 212 |
| 2 | (H, 0.078) | (L, −0.002) | 168 |

Figure 14 depicts a graphical representation of the obtained clusters, Cluster 0 (green), Cluster 1 (dark purple), Cluster 2 (yellow).

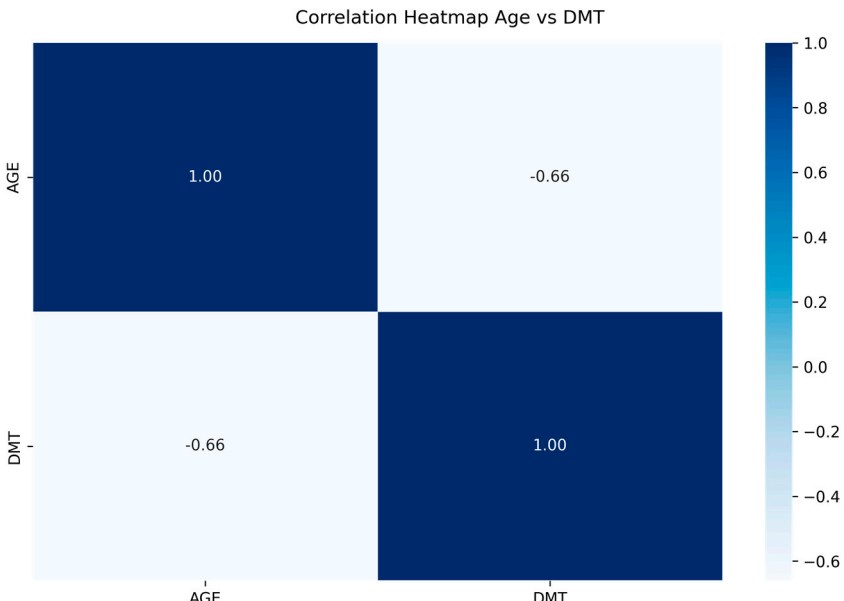

**Figure 13.** Correlation matrix.

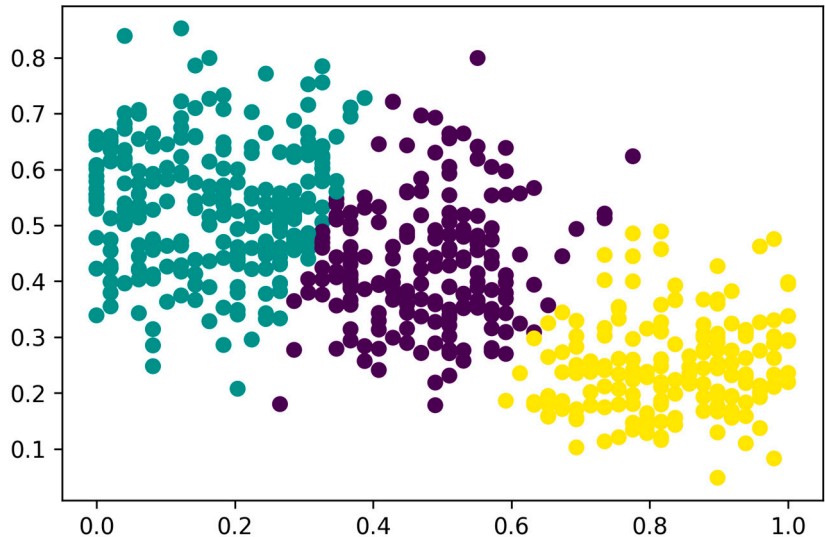

**Figure 14.** Cluster of digital maturity of tourists vs. age.

## 6. Discussion

Through the implemented process, individual assessment scores can be generated, reflecting the digital maturity level of tourists in a specific Smart City. The model incorporates a methodology that combines decision theory using the AHP and fuzzy logic. This approach allows for the formulation of weights for each variable in the model, facilitating the customization of the assignment process, the inclusion of new criteria, and enhancing decision making through the involvement of multiple experts belonging to public and private entities [56]. The model's adaptability ensures its capability to be tailored to specific requirements and to accommodate changes in the technological landscape within the Smart City environment.

The implementation of the tourist clustering process involved the application of the k-means algorithm, which was chosen due to its effectiveness in partitioning data into distinct clusters based on their similarities. The primary goal was to group tourists with similar characteristics to gain insights into their digital maturity levels and preferences for a more personalized tourist experience in Smart Cities.

During the data collection phase, anonymous visitor data were obtained from various tour operators operating in the city. The data encompassed a diverse range of variables, including age groups and the criteria used in the definition of the digital maturity level. Notably, a strong correlation was observed between the age groups and the level of digital maturity, indicating that certain age segments tended to exhibit higher digital literacy and engagement with smart technologies.

The k-means algorithm was employed to carry out the clustering process, aiming to identify meaningful patterns and segments within the dataset. Through multiple iterations, the k-means algorithm partitioned the 600 tourists into distinct clusters, each representing a specific profile based on their age and digital maturity. The clustering process allowed for a deeper understanding of the preferences and behaviors of tourists within each group.

As a result of this clustering, practical implications emerged for Smart Cities' tourism strategies. For instance, the identification of age-specific clusters with varying digital maturity levels enabled the city to customize and optimize its tourist offerings. Younger and more digitally savvy tourists were provided with advanced smart services, while older tourists were offered more user-friendly, technology-enhanced experiences.

The obtained scores serve to classify each tourist individually, enabling the creation of Customer Journeys tailored to their technological background, adaptability to technology, and the integration of technology within Smart Cities. By utilizing the clusters defined in Table 7, users can be categorized into three distinct groups based on their technology usage within the Smart City environment. Once a person is assigned to a specific profile, it becomes relatively straightforward to generate interactions that make the tourist feel welcomed throughout their stay in a Smart City. In Table 8, recommendation strategies are provided for each cluster.

**Table 8.** Recommendation strategy for each cluster.

| Cluster c | Recommendation Strategy |
|---|---|
| 0 | Description: This cluster corresponds to tourists with a moderate level of digital maturity. The age range falls within the middle age range (46–55 years), indicating tourists with a positive ability to adapt to technology and, consequently, potential users of the technology associated with Smart Cities. Recommendation: This type of tourist can be guided throughout their visit by utilizing mobile technology as a key component of their digital journey. Primarily, a chatbot can be employed as a fundamental tool to provide recommendations based on the tourist's geographic location and personal preferences, which can be obtained through a prior survey. |
| 1 | Description: This cluster corresponds to tourists with a moderate level of digital maturity. The age range falls within a lower age range, specifically 26–45 years. Similar to the previous cluster, these tourists exhibit a positive ability to adapt to technology and therefore have the potential to be users of the technology associated with Smart Cities. This younger age group implies greater possibilities for interactions and recommendations for guided visits. Recommendation: In this case, similar to the first cluster, mobile technology can be used as a fundamental component in the process of communication and building personalized Customer Journeys. Additionally, in this case, activities can be recommended through the use of chatbots, offering digital challenges to enhance the tourist's digital journey. Furthermore, it is possible to facilitate group interactions with other individuals who are interested, creating personalized and focused digital experiences centered around the achievement of challenges and interactions with similar profiles. The use of mobile technology and geolocation enables real-time individual and group recommendations. |
| 2 | Description: This final cluster corresponds to individuals aged 56 and older with a low level of digital literacy. Recommendation: For this segment, it is recommended to prioritize personalization through human interaction, leveraging factors such as geolocation and planning activities based on individual preferences. Additionally, organizing group activities with others in the same cluster who share similar interests and preferences is suggested. |

Although in each cluster personalized recommendations are made according to the level of digital maturity, personalized service models can be developed according to the tourist's evaluation and other factors, such as their recommendations, the use of Smart

City technology, the capacity to influence social networks, etc. Basically, working on the following lines of action:

- Offering personalized services and experiences.
- Targeting the promotion of events and activities.
- Personalizing communication and recommendations.
- Enhancing tourist satisfaction and retention.
- Collecting data for future improvements.

While the classification into clusters is primarily influenced by the correlation between age and DMT, it is important to acknowledge that individual characteristics and preferences may vary within each age group. It is indeed possible for older individuals to exhibit a high level of digitization and, consequently, be assigned to a different cluster. Likewise, younger individuals may require personalization in recommendations despite their familiarity with technology. At this stage, the DMT classification governs the behavior of the clustering, guiding the development of these relationships. It is crucial to consider individual variations, further refine the clustering approach to account for these nuances, and better tailor the digital experiences and recommendations for each tourist.

The average assessment of digital maturity as a function of age is shown in Figure 15.

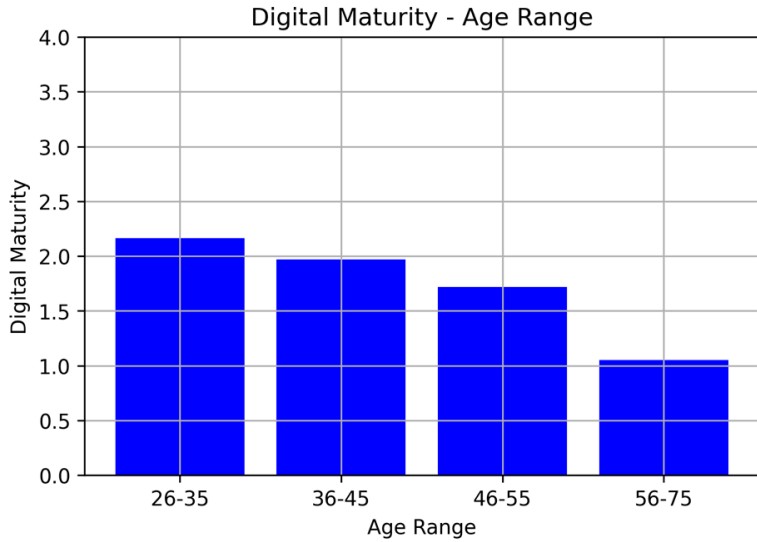

**Figure 15.** Bar chart of digital maturity of tourists vs. age.

Evidently, there is an inverse relationship between age range and the average level of digital maturity. As the age range increases, the average level of digital maturity tends to decrease, and vice versa. This observation aligns with the common trend where younger individuals tend to have a higher familiarity and comfort level with digital technologies compared to older individuals. It highlights the importance of considering age as a significant factor when assessing and addressing digital maturity levels in the context of Smart Tourism.

## 7. Conclusions

This study aimed to develop a tourist evaluation system within the context of a Smart City environment. A comprehensive working methodology was proposed, which can be expanded to incorporate a wider range of criteria, alternatives, and multiple decision makers. The methodology leveraged the analytic hierarchy process (AHP) as a decision-making tool and utilized fuzzy logic to establish a linguistic model with minimal information loss. Furthermore, a clustering process was applied in a subsequent phase to provide tailored global recommendations based on specific tourist clusters and segments, considering their level of digitization. This entire model contributes to the development of personalized

communication processes in the relationship between customers (tourists) and public or private entities.

The results of the study revealed that the digital maturity level of the examined sample varied from low to medium. This finding emphasizes the need for targeted training programs focused on enhancing the perceived usefulness and ease of use of technology applications within the tourism sector. It is crucial to capitalize on the potential benefits that Smart Cities can offer to the tourism industry. Additionally, an essential observation is the significance of technological competency and the attitudes towards technology among individuals. These factors must be supported by strategies that enhance the perception of the usefulness of technological applications and subsequently facilitate their ease of use.

It is worth noting that the developed model is not limited to the tourism sector; it can be extended to other sectors such as industry, finance, environment, sustainability, or any other applicable use. The defined clusters exhibit a strong interrelation with each other, emphasizing the inherent correlations within the data.

Furthermore, fostering collaboration between the public and private sectors is essential to effectively communicate the importance of digitalization and the transformative benefits that the implementation of Smart City technologies brings to citizens, specifically within the tourism sector. By working together, these sectors can jointly promote and advocate for the adoption of digital solutions, emphasizing how these advancements enhance the overall tourist experience and contribute to the sustainable growth of cities.

To operationalize the recommendations derived from the identified clusters, a multi-faceted approach is required. Apart from focusing on personalized interactions between tourists and public or private entities, it is crucial to implement a customer-centric technological infrastructure. This involves deploying innovative digital tools, such as mobile applications, augmented reality guides, or interactive maps, which facilitate seamless communication, real-time information sharing, and personalized service delivery. By leveraging these technologies, tourists can enjoy customized experiences that align with their preferences, interests, and digital capabilities.

However, the successful implementation of such technological initiatives goes beyond deploying the systems alone. It necessitates a comprehensive and tailored training program that equips individuals with the necessary digital skills and knowledge. This training should be designed to address the varying levels of digital maturity and the different age groups within the target audience. By empowering citizens with the digital competencies needed to fully embrace and utilize the available technologies, cities can ensure inclusivity and maximize the benefits of Smart Tourism.

Furthermore, continuous evaluation and feedback loops should be established to monitor the effectiveness of the digitalization efforts and identify areas for improvement. By gathering insights from tourists, analyzing their preferences, and adapting the digital offerings accordingly, both the public and private sectors can refine their strategies, enhance the tourist experience, and foster a culture of continuous innovation.

In summary, a collaborative approach between the public and private sectors is essential to promote digitalization and the adoption of Smart City technologies in the tourism sector. By prioritizing personalized experiences, leveraging advanced technologies, and providing tailored training programs, cities can create a digital ecosystem that enhances the overall tourist experience, fosters sustainable growth, and empowers citizens to fully embrace the digital transformation journey.

The conclusions drawn can be expanded and validated through the following analyses:

- Investigating emerging trends or advancements in Smart Cities and Smart Tourism can help the study to stay current and provide novel insights into how new technologies and practices can further enhance the tourist experience.
- Examining how tourist preferences and behaviors evolve over time can offer valuable information on the sustainability and lasting impact of Smart Tourism strategies.

- Comparing the effectiveness of various Smart Tourism strategies and technologies across different contexts means that the study can provide nuanced recommendations tailored to specific settings.
- Exploring the perspectives of various stakeholders, such as tourists, city officials, and businesses, can shed light on the multi-dimensional aspects of Smart Tourism implementation.
- Investigating the policy and regulatory implications of Smart City and Smart Tourism initiatives.

## 8. Future Works

From the conclusions of this study, which integrate bibliographical knowledge, understanding, and expansion of the TAM model, as well as the measurement of customer value in relation to the use of technology applied to Smart Cities, several areas for improvement and potential avenues for future research can be highlighted. These include the following:

- Investigating the impact of digital maturity on the effectiveness of Smart City technologies: Further studies can delve deeper into understanding how the level of digital maturity influences the adoption and utilization of Smart City technologies among citizens. Examining the relationship between digital skills, attitudes, and technology acceptance can provide valuable insights into tailoring strategies to bridge the digital divide and enhance engagement.
- Designing personalized digital experiences for tourists in Smart Cities: Future research can focus on developing innovative approaches to create highly personalized digital experiences for tourists. This can involve leveraging emerging technologies such as artificial intelligence, machine learning, interpretable decision making [57], and personalized recommendation systems to deliver customized recommendations, interactive itineraries, and immersive digital content based on individual preferences and needs.
- Evaluating the long-term impacts of Smart Tourism initiatives: Longitudinal studies can be conducted to assess the long-term effects of implementing Smart Tourism initiatives within Smart Cities. This can involve analyzing the economic, social, and environmental outcomes of digitalization efforts, including the sustainability of Smart Tourism practices and their influence on local communities, cultural heritage, and the overall tourist experience.
- Examining the role of data privacy and security in Smart Tourism: As the collection and utilization of personal data become integral to Smart Tourism practices, future research can explore the ethical and legal considerations surrounding data privacy, security, and consent. Investigating ways to ensure transparency, trust, and data protection in Smart Tourism initiatives can help address privacy concerns and foster a positive perception of the digitalization efforts. By incorporating ethical considerations into future works, the tourism industry can proactively address concerns about the ethical use of data, ensuring tourists' privacy is respected, and their personal information is secure. By fostering an ethical approach, Smart Cities can build trust with tourists, enhancing their overall satisfaction and promoting the success and sustainability of Smart Tourism initiatives.
- Assessing the scalability and transferability of the proposed model: The applicability and effectiveness of the developed model can be further evaluated across different Smart City contexts and sectors. Conducting comparative studies in various geographic locations and industries can provide insights into the adaptability and transferability of the model, as well as allowing the identification of sector-specific variations in digital maturity and criteria relevance.
- Examining the impact of Smart Tourism on destination competitiveness: Future research can focus on exploring how the adoption of Smart Tourism practices influences the competitive advantage of destinations. Evaluating the economic and strategic implications of Smart City technologies on destination branding, marketing strategies,



and visitor satisfaction can help guide decision making and investment in Smart Tourism initiatives.

These future research directions aim to advance the understanding of the evolving landscape of Smart Tourism within Smart Cities and to contribute to the development of effective strategies, frameworks, and technologies to enhance the digital experiences of tourists and maximize the benefits of digital transformation in the tourism sector.

**Author Contributions:** Conceptualization, G.M.D. and J.L.G.S.; methodology, G.M.D.; software, G.M.D.; validation, G.M.D., J.L.G.S. and J.J.G.H.; formal analysis, G.M.D. and J.L.G.S.; investigation, G.M.D., J.L.G.S. and J.J.G.H.; resources, G.M.D., J.L.G.S. and J.J.G.H.; data curation, G.M.D.; writing—original draft preparation, G.M.D. and J.L.G.S.; writing—review and editing, G.M.D., J.L.G.S. and J.J.G.H.; visualization, G.M.D.; supervision, G.M.D., J.L.G.S. and J.J.G.H.; project administration, G.M.D.; funding acquisition. All authors have read and agreed to the published version of the manuscript.

**Funding:** This research received no external funding.

**Data Availability Statement:** Not applicable.

**Conflicts of Interest:** The authors declare no conflict of interest.

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
