# Peer review of "Smart Cities and Citizen Adoption: Exploring Tourist Digital Maturity for Personalizing Recommendations"

_electronics, doi:10.3390/electronics12163395_

Round 1
Reviewer 1 Report
The article titled "Are Smart Cities Successfully Accepted by all Citizens? A New Recommender System Applied to Tourist Mobile Technologies" explores the concept of Smart Cities and Smart Tourism. The study proposes a novel recommender system to enhance the tourist experience that combines the Technology Acceptance Model (TAM) with the fuzzy 2-tuple linguistic model and the analytic hierarchy process (AHP). Although the research addresses an important topic, it requires major revisions to strengthen its content and contribution. Here are the major points that need improvement:
The current title is quite long and can be shortened to convey the main focus of the study more effectively. The revised title emphasizes "Citizen Acceptance of Smart Cities" as the primary theme. Additionally, since there is no specific "Recommender System" algorithm used in the proposed methodology, it would be more accurate to use "Recommendations" instead.
The introduction starts by highlighting the constant changes and challenges faced by the world due to economic growth and new technologies. It would be beneficial to provide more specific examples or statistics to illustrate the increasing urbanization and the impact of technology on cities. This will help readers understand the urgency and relevance of the topic.
The authors mention the lack of studies on citizen acceptance of Smart Cities and Smart Tourism technologies, but it could further emphasize the significance of this research gap. Explain how this gap affects the successful implementation and sustainability of Smart Cities and Smart Tourism initiatives. This will make the motivation for the study clearer.
Provide concrete examples of how technologies employed in Smart Tourism can contribute to the overall intelligence and efficiency of a Smart City. Discuss how successful adoption of new technologies by citizens can lead to improved services, enhanced urban experiences, and sustainable development. This will further emphasize the significance of the research paper.
At the beginning of the methodology section, the authors provide a concise summary of the objectives and purpose of the research. Clearly state how the integration of AHP and LD2T will contribute to the proposed research and its relevance in the context of Smart Cities and Smart Tourism. While the section discusses the 2-tuple linguistic model (LD2T) in detail, it would be beneficial to provide a similar level of explanation for the analytic hierarchy process (AHP). Describe the AHP methodology, its principles, and how it aids in decision-making. Additionally, explain the rationale behind choosing AHP for this study and its compatibility with the fuzzy linguistic approach.
Integration of AHP and LD2T: Provide a more explicit explanation of how AHP and LD2T are integrated to form the proposed model. Clarify how AHP structures the decision-making process and how LD2T complements it to handle heterogeneous data. This will help readers understand the synergy between the two models.
Explain in greater detail why the fuzzy linguistic model is necessary for handling heterogeneous data in the specific context of Smart Tourism and Smart Cities. Describe how it captures uncertainties and preferences and how it enhances the precision of information representation for effective decision-making. To enhance understanding, include practical examples or scenarios that illustrate the application of AHP and LD2T in the context of Smart Tourism and Smart Cities. Demonstrate how these models aid in prioritization, personalized interactions, and improving the tourist experience. Explain how this approach outperforms other alternatives and contributes to the novelty of the research. Additionally, the authors should elaborate on the data collection process, including the sources of data and the criteria used to evaluate citizen acceptance. Providing transparency about the data collection will strengthen the study's credibility and allow readers to assess the generalizability of the findings. By incorporating these improvements, the methodology section will offer a more comprehensive and coherent understanding of the research approach, further strengthening the credibility of the study.
The discussion section effectively presents the implementation process of the tourist evaluation system and its classification into three clusters based on digital maturity. However, it can be enhanced by including more specific details about the methodologies used in the implementation, the data collection process, and the clustering algorithm. The discussion on the cluster recommendation strategy is informative, providing tailored approaches for each cluster. To further improve this section, consider discussing potential challenges or limitations in implementing these strategies and how they might be addressed. Discuss the practical implications of the clustering results. How can the identified clusters be utilized to improve the overall tourist experience in Smart Cities? Explain how personalized communication and recommendations can be tailored based on the cluster profiles to enhance tourist satisfaction.
Emphasize how addressing the research directions can further validate or expand upon the conclusions and recommendations provided in the study.
Consider adding contents in the future works about addressing ethical concerns related to smart tourism and technology adoption. Discuss the importance of data privacy, security, and responsible use of digital technologies to build trust with tourists and enhance the success of smart tourism initiatives.
Overall, the quality of the English language in the article is good, with coherent and well-structured writing. However, some areas can be improved, such as sentence structure, word choice, grammar, and clarity. The title could be shortened for better readability. Careful proofreading and editing will enhance the overall quality of the paper.
Author Response
Thank you for all the enriching comments; they have greatly helped us to guide and focus the article more appropriately. We have changed the title to the following: "Smart Cities and Citizen Adoption: Exploring Tourist Digital Maturity for Personalizing Recommendations," as we believe it aligns better with the article's objectives.
Additionally, in the attached document, you will find the responses to all the questions raised.
Best regards.

Reviewer 2 Report
A detailed review was attached.

Typos and grammatical mistakes, especially punctuation mistakes.
Author Response

(The authors gave the same response as above.)

Reviewer 3 Report
Overall the paper presentation is appropriate. However, the authors need to modify the paper before publication
1- Abstract should be up to 10 lines maximum.
2- Summarize objectives in 3 points.
3- When introducing related work, it is important to focus on the literature related
to smart cities. However, the paper only mentions a few related literatures. The specific contributions or findings are not highlighted in detail. Also, you need to investigate works like in
https://ieeexplore.ieee.org/abstract/document/8782301 from different points of view.
4- Please add a flowchart and related discussion in Section 3.
5- There are obvious errors in formula 7, which needs verification.
5- The results were poorly discussed in light of the previous research findings. Please put results in the context of what was known before and then discuss potential challenges, future directions/recommendations, and why these kinds of studies are important!
6- Overall the English and the paper organization must be improved.
Moderate
Author Response

(The authors gave the same response as above.)

Reviewer 4 Report
1. What is the main question addressed by the research?
The aim of the paper is presenting a new model of recommender system based in the most cited and used model in the scientific and academic literature: the Technology Acceptance Model (TAM) and using the most cited criteria in the vast literature. This goal has been fully achieved.
2. Do you consider the topic original or relevant in the field? Does it
address a specific gap in the field?
The purpose of the article is important because the development of smart urban solutions undoubtedly depends on their acceptance by users. The proposed model is interesting and original, although the conclusions obtained are quite obvious.
3. What does it add to the subject area compared with other published
material?
The novelty of the results consists in: embedding research in tourism, using advanced econometric methodology, integrating existing approaches and verifying it in urban reality.
4. What specific improvements should the authors consider regarding the
methodology? What further controls should be considered?
The methodology based on multi-criteria analysis and fuzzy logic is well chosen and presented. I have no comments.
5. Are the conclusions consistent with the evidence and arguments presented
and do they address the main question posed?
The conclusions are synthetic but exhaustive. They do not require changes.
6. Are the references appropriate?
The bibliography could be wider, but due to the industry-specific considerations, it can be considered sufficient.
7. Please include any additional comments on the tables and figures.
The graphic material is extensive and very well developed. It perfectly illustrates the considerations.
Author Response

(The authors gave the same response as above.)

Round 2
Reviewer 1 Report
After reviewing the authors' response to the comments, we find that they have adequately addressed all concerns. The revisions have improved the article significantly, and we recommend its acceptance.
English language and style are fine/minor spell check required.
Reviewer 2 Report
It can be accepted with this version.